# How IoT-Driven Citizen Science Coupled with Data Satisficing Can Promote Deep Citizen Science

**DOI:** 10.3390/s22093196

**Published:** 2022-04-21

**Authors:** Stefan Poslad, Tayyaba Irum, Patricia Charlton, Rafia Mumtaz, Muhammad Azam, Hassan Zaidi, Christothea Herodotou, Guangxia Yu, Fesal Toosy

**Affiliations:** 1School of Computer Science and Electronic Engineering, Queen Mary University of London (QMUL), London E1 4NS, UK; t.irum@qmul.ac.uk (T.I.); guangxia.yu@qmul.ac.uk (G.Y.); 2School Computing and Communication, Institute of Educational Technology, Open University (OU), Milton Keynes MK7 6AA, UK; patricia.charlton@open.ac.uk (P.C.); christothea.herodotou@open.ac.uk (C.H.); 3School of Electrical Engineering and Computer Science, National University of Sciences and Technology (NUST), Islamabad 44000, Pakistan; rafia.mumtaz@seecs.edu.pk (R.M.); drzaidi@seecs.edu.pk (H.Z.); 4School of Information Technology, Whitecliffe College, Auckland 1010, New Zealand; muhammada@whitecliffe.ac.nz; 5Faculty of Engineering, University of Central Punjab (UCP), Lahore 54000, Pakistan; fesal@ucp.edu.pk

**Keywords:** Internet of Things (IoT), citizen science (CS), data quality, data satisficing

## Abstract

To study and understand the importance of Internet of Things-driven citizen science (IoT-CS) combined with data satisficing, we set up and undertook a citizen science experiment for air quality (AQ) in four Pakistan cities using twenty-one volunteers. We used quantitative methods to analyse the AQ data. Three research questions (RQ) were posed as follows: Which factors affect CS IoT-CS AQ data quality (RQ1)? How can we make science more inclusive by dealing with the lack of scientists, training and high-quality equipment (RQ2)? Can a lack of calibrated data readings be overcome to yield otherwise useful results for IoT-CS AQ data analysis (RQ3)? To address RQ1, an analysis of related work revealed that multiple causal factors exist. Good practice guidelines were adopted to promote higher data quality in CS studies. Additionally, we also proposed a classification of CS instruments to help better understand the data quality challenges. To answer RQ2, user engagement workshops were undertaken as an effective method to make CS more inclusive and also to train users to operate IoT-CS AQ devices more understandably. To address RQ3, it was proposed that a more feasible objective is that citizens leverage data satisficing such that AQ measurements can detect relevant local variations. Additionally, we proposed several recommendations. Our top recommendations are that: a deep (citizen) science approach should be fostered to support a more inclusive, knowledgeable application of science en masse for the greater good; It may not be useful or feasible to cross-check measurements from cheaper versus more expensive calibrated instrument sensors in situ. Hence, data satisficing may be more feasible; additional cross-checks that go beyond checking if co-located low-cost and calibrated AQ measurements correlate under equivalent conditions should be leveraged.

## 1. Introduction

Science seeks to objectively understand and explain the events of nature in a reproducible way. As noted by Silvertown [1], the first scientists were not paid professionals but amateur scientists, an early type of citizen scientist. Currently, modern professional scientists to date carry out the vast majority of scientific research and development. However, since the mid-1990s, a new form of amateur citizen science (CS) has been proposed by Irwin [2] that works in collaboration with, yet is usually led by, professional scientists, such that: first, science can be responsive to citizens’ concerns and needs; and second, that citizens themselves could produce reliable scientific knowledge.

According to Cooper and Lewenstein [3], there are two different facets to Irwin’s definition [2]. The first emphasises the responsibility of science to society, which is termed “democratic” citizen science. The second facet is “participatory” citizen science, as a practice in which people mostly contribute observations or efforts to the scientific enterprise. A comprehensive review of CS terminology is given in [4]. However, the term CS is commonly attributed to Irvin in 1995 as one of the first researchers to more clearly explain and explicitly promote it. Vohland et al. [5] state that the term CS was used even earlier in 1989 by the Massachusetts Institute of Technology (MIT). The author of [5] also contends that CS needs to encompass and promote an open and broad understanding of multiple research practices and participatory activities that take place when people who are not tasked with carrying out research as part of their paid work get involved in it. They also assert multiple definitions are essential for the development of CS, including its enabling frameworks, mechanisms, and the different needs of specific applications.

Pelacho et al. [6] raise the issue of who governs and manages the fruits of labour resulting from CS. It often seems that it is more professional scientists, authorities and businesses rather than citizens that benefit most from CS. Pelacho et al. [6] propose to manage science as a commons, a form of community management of a shared resource. The commons results from a collaborative, open, and experimental process that necessarily involves a community of practice as a group of people who share a concern or a passion for something they do and learn how to do it better when they interact regularly.

There are different drivers for citizen science, including but not limited to:Aid scientists through *citizens by providing wider, more temporal-spatial measurements* of the physical and natural world than scientists’ conventional approaches, e.g., using mobile air quality monitoring rather than that based upon networks of static, sparse measurement stations [7];*Environment monitoring equipment may be prohibitively expensive* to be used to capture temporal-spatial heterogeneity and identify environment hotspots leading to the development of robust real-time strategies for exposure control [8]—hence the need for low-cost sensors (LCS) such as IoT-based approaches;The *lack of scientists and specialised resources* to monitor the physical and natural world, especially in developing countries [6];Serve the education and outreach goals *to widen, make more inclusive citizens’ participation in science* [9].

Whilst some applications of citizen science can be performed without any technology, i.e., through using human observation, technology has become an important enabler for citizen science [1], including the use of an Internet of Things (IoT), where a wider range of environment measurement devices that includes low cost and low resource devices can be used to instrument, sense and exchange data about the environment. In this paper, we focus on a subset of CS, on the use of IoT-driven or IoT-enabled CS (IoT-CS).

A further focus is on the challenge of producing good quality data through IoT-CS. Although data quality is a well-known general concern for the CS, much of the analysis of data quality issues for CS is quite high-level and quite generic, in contrast to the focus here. It is often taken for granted that CS data quality should attain the level of professional scientist data quality, e.g., to ascertain how well the sensed state of the environment meets regularity standards or norms [3]. In brief, (see the related work in Section 2 for more details), there is a range of reasons why this is somewhat unrealistic for specific types of CS to achieve, such as IoT-CS. This includes a lack of CS training and the use of lower quality equipment; measurements may occur in more varied less-defined conditions that affect instruments’ operations.

So, if the main decision or outcome by scientists is the CS data quality is such that it can validate the state of the environment being normal or not to meet regulatory concerns, this can be hard to achieve by CS [10]. This aim may be far less of a concern for CS than for professional scientists. Thus, we introduce the concept of satisficing, a decision-making strategy that entails searching through the available alternatives until an acceptability threshold is met [11]. For CS, a relevant optimal solution may not be able to be determined because many natural problems are characterised by computational intractability or a lack of information. Hence, the main novel contributions of this paper are:1.Analysis and classification of surveyed CS projects, especially with respect to the CS equipment used.2.Analysis of IoT-enabled CS issues, in part based upon a field study to determine AQ across four different cities in Pakistan.3.Recommendations for IoT-enabled CS, data satisficing and deep CS.

It is vitally important to monitor the physical environment scientifically, with a focus on certain air pollutants that are relevant to both industrial and residential areas. In a country like Pakistan, where a large part of the population is concentrated around the largest cities, the density of air pollutants can reach unhealthy to hazardous levels. This is partly due to the tropical climate, particularly in the cities where this study was conducted. These cities experience very little wind and rain in autumn and winter, and due to a temperature inversion in November, various pollutants mix with the atmospheric humidity creating a layer of thick ‘smog’ that covers the entire city [12]. There are too few scientists in the world, let alone in a low/middle-income country like Pakistan that have the skill required to monitor and study these conditions. In these circumstances, it would be helpful to motivate and equip citizens to join in with the monitoring and data collection process. This would greatly improve the temporal–spatial density of the data and would also help create much-needed awareness about the importance of air quality and pollutants in the atmosphere.

To study and understand the importance of IoT-driven citizen science (IoT-CS) combined with data satisficing, three research questions (RQ) were posed in this study as follows:
RQ1—Which factors affect CS IoT-CS data quality in the context of AQ?RQ2—How can we make science more inclusive by dealing with the lack of scientists, training and high-quality equipment through deploying IoT-CS?RQ3—Can a lack of calibrated data readings yield otherwise useful results for IoT-CS AQ data analysis?

The remainder of this paper is organised as follows. The next section, Section 2, surveys related work. Section 3 presents our method and results. Finally, Section 4 presents the conclusions and recommendations.

## 2. Related Work

The focus of this survey is as follows: To highlight the challenges of using LCS IoT devices as scientific instruments to undertake CS applied to AQ monitoring; to better understand what we can do when the resulting data quality of CS is variable; to more appropriately frame how this affects the decisions about its impact and the implications of its outcomes. It is well-known that a major challenge with CS is that amateur citizens can produce lower quality data than professional scientists. Rather than classify and analyse CS projects as a single group or by science topic, the equipment used for CS and the challenges in using LCS IoT equipment for CS to generate quality data were considered. Hence, we first postulated a taxonomy of CS equipment, as shown in Figure 1.

Figure 1 describes three basic types of equipment used for CS: humans as sensors or no instrument human observations, LCS versus standard scientific instruments. Another dimension is that these range from simple to Do It Yourself (DIY) IoT instrument creation to the use of standard, calibrated, scientific instruments. A further dimension is the use of amateur or less-well trained instrument operators versus trained or scientific operators. This is in contrast to a coarse-grained classification of data quality errors by human participants or from the protocols being used for CS [13]. This taxonomy highlights that different types of CS instruments can introduce different challenges, including those for data quality that govern the outcomes and decisions that can be drawn from that data. Projects employing IoT systems of sensors for monitoring the environment require a quantitative analysis of uncertainties for data quality estimates. While analysing the data quality of projects on environmental monitoring, environmental factors or operating conditions also need to be considered as these can impact the performance and efficiency of the sensors. The focus of this survey is on LCS IoT-CS systems and deployment issues therefrom and data quality.

If we classify CS projects with respect to the equipment they use, the subset of IoT-CS projects tends to have a specific set of additional challenges, e.g., as surveyed in part by Kumar et al. for AQ IoT projects [8]:*Sensor calibration and reliability*: Some cheap sensors have no defined calibration to relate their given voltage outputs to measurement units. Measurements may need to be evaluated under a range of ambient environmental conditions, e.g., temperature and humidity, as these affect the AQ measurements. Hence, cross-checks of LCS with calibrated instruments in the lab or under only a few field conditions need to be considered where feasible. Relevant metrics for the comparison of LCS systems against reference systems should be considered. Ref. [14] highlighted the most cost-effective LCS that could be used to monitor air quality pollutants versus reference systems with a good level of agreement represented by a coefficient of determination R^2^ > 0.75 and a linear slope of the regression line between these, close to 1.0. Note that R^2^ is overly dependent on the range of reference measurements, the duration of the test, and the season and location of the tests, meaning that changes in R^2^ are not completely dependent on LCS or their calibration methods [15]. Note, also that citizens would also need a business, local authority or scientific partner that has access to a calibrated instrument, as these are expensive to buy. Periodic recalibration may also be needed as the sensitivity of sensors can change over time.*Sensor sensitivity versus selectivity:* A limitation in improving the sensitivity of the AQ sensors is that some (AQ) sensors may be marketed for one pollutant but exhibit cross-sensitivity to other pollutants or contaminants. This is especially an issue for gases at low concentrations, in the parts per billion (ppb) range. Such contaminants can contribute to a biased response of the sensors, thereby deteriorating their selectivity.*Sensor stability and response time*: Sensing (AQ) mechanisms can involve quite complex chemical reactions, e.g., oxidation and reduction of the sensing materials and adsorption of oxygen and other chemical species on their surface, and catalytic reactions between the adsorbents. As a result, the performance of these sensors is sensitive to their operating conditions. A waiting time may be needed for sensors to stabilise, and if the conditions change, i.e., sensors are moved, further stabilisation times are needed before reliable measurements can be undertaken.*Slow changes in environment state*: The physical world often changes slowly, from minutes to hours to days or even longer, in part because of its mass, leading to biological, chemical and physical phenomena that also change slowly, including human health effects. Short periods of measurement do not capture such changes. Measurement equipment may not be able to be safely left and maintained (powered up) to attain such measurements. Note also that some standardised measurements for the USA AQ Index (AQI) require taking long periods of measurements from several hours to a day to get a standard average value [16]. It is also difficult to quantitively ascertain that isolated short exposure periods of less than a day to some potentially harmful environment states are, in turn, harmful to humans and other life forms too.*Variable operating conditions*: e.g., temperature and humidity, can affect the operation of (AQ) sensors, and these vary temporally and spatially [17]. These may also be affected by artificial hot spots, e.g., machines generating outbursts of hot or humid air leading to noisy data. Hence such operating conditions should also be recorded too. These conditions include location coordinates, location context as the type of location, indoors versus outdoors, proximity to any suspected pollution sensors, the height of sensors above ground level, if and how the human holding the sensor is moving or not when sensing, time and weather conditions.*Resilience*: IoT sensor systems need to be designed to operate reliably under a range of diverse environmental conditions and to be moved.*Equipment costs:* the higher the IoT-CS equipment cost, the more exclusive the group of citizens that can afford to procure it to use, perhaps excluding low-income neighbourhoods that may be more at risk of the effects of poor physical environment quality.*Maintenance or operational costs*: These may be considerable, e.g., for calibration, battery recharging or replacement, data management, analysis, and visualisation. The maintenance costs may easily exceed the cost of the actual IoT CS equipment itself.*Longevity* of components before replacement. Note, AQ sensors could have a working time of the order of six months to a couple of years at most.*Trust* by the scientific community and decision-makers as they may not be prepared to embrace such technology.*Data quality* of the monitored phenomena needs to be addressed effectively and transparently to maximise the benefit to scientists, the public and authorities to help make more informed decisions. However, it may not be simple to just automatically collect and publish the data. This is because of the issues concerning:oDifferences in conditions of context, such as location [15], time, temperature and humidity, between different instruments’ measurements.o*Labelling* the data with the measurement operating conditions (measurement context) to help understand the meaning of the data;o*Pre-processing or filtering* the data before publishing;o*Standardising* the data in terms of common measurement units, data format, etc.
*Data dissemination* issues of the monitored phenomena need to be addressed effectively and transparently:o*Anonymising* any personal context for the data;o*Visualising and providing understandable commentaries* or interpretations of the data to citizens.


The focal point of this study is first to analyse the limitations of how to improve scientific instrument use and the quality of the data-driven science within IoT orientated CS projects applied to AQ studies. In addition, we selectively analyse applications somewhat wider than AQ studies to discuss the wider issues of using data-driven science within IoT orientated CS projects for deep CS.

In the CitiSense project [18], an AQ IoT-CS that sensed carbon monoxide (CO), nitrogen dioxide (NO_2_), and ozone (O_3_) was trialled by 16 mobile commuters. The main objective was to investigate the temporal-spatial distribution of standardised readings of AQ to enable the identification of pollution hot spots and microenvironments. Kriging was used to interpolate incomplete data. The main finding was that CitiSense could provide more details about the AQ distribution than that published by a standard environment agency. Data quality issues were not discussed much. However, sensor calibration was performed in a Lab, and the need for periodic calibration was noted.

In [19], an AQ IoT-CS consisting of multiple microcontroller unit (MCU) boards was combined with LCS to sense dust concentration (as an indicator of air quality), noise, temperature, humidity, and pedestrian count as citizens walked along roads in a city. The main findings were that: more citizen training could improve the reliability and accuracy of the CS sensor data collected; context data about key events that affect the data quality would be useful to be recorded to provide a linkage analysis of cause to effect.

In [20], the focus was on mobile AQ sensing based upon an Arduino MCU board that is integrated with sensors to measure CO (Volatile Organic Compounds (VOC), O_3_, NO_2_, gasoline and diesel exhaust, temperature, humidity and location via GNSS (Global Navigation Satellite Systems). This considered that cross-contamination and changing environment conditions (temperature and humidity) affect the gas concentration. It also enclosed the measurement device in a box with a regulated airflow to mitigate against movement changing the airflow and gas density. They considered the limited nature of in lab sensor calibration and considered infield calibration through cross-comparing multiple sensors’ readings.

Some researchers have focused in particular on three main techniques for LCS calibration: sending a calibrated measurement value or signal periodically to an in-field sensor (operating in similar conditions); requiring citizens to self-calibrate sensors using chemicals (gases) at known concentrations; co-locating LCS with a standard reference calibrated device, and then cross-comparing measurements [21]. A few studies have compared co-located LCS against calibrated instruments, e.g., Castell et al. [22] compared multiple instances of an LCS IoT AQ device against a calibrated device. There was a reasonable agreement for NO and particles on the order of 2.5 μm or less (PM2.5), but less so for NO_2_, O_3_, PM_10_ and CO. Another interesting finding was the response of different instances of the same type of IoT AQ varied concerning changing operating or environmental conditions. Ref. [23] undertook a comparison of a PM LCS to a calibrated one and noted that the agreement between the two varied concerning particle size.

Some other IoT AQ platforms or projects give very little technical information in published papers about the components used, such as sensors, the data acquisition board, data quality issues, sensor calibration, etc., e.g., OpenSense [24] and ExposureSense [25].

Although the focus of this paper is predominantly on CS combined with IoT applied to the AQ domain, rather than on a survey of CS in general, we nevertheless highlight some other CS work that comments on equipment usage, operation and data quality issues. Others specify the equipment but say little about data quality, e.g., [26,27] proposed that user studies are fully-integrated into CS to aid technological implementations. However, these papers say little about the quality of the science, sensors used and data. In contrast, Ref. [28] considers how data quality could be improved in CS projects, albeit applied to weather rather than AQ measurements. They propose good practice guidelines concerning six aspects such as (a) giving equipment operational instructions in an accessible language, avoiding technical terminology; (b) providing technical details about different devices and their accuracy; (c) asking citizens to report contextual information such as the instrument they used and the operational conditions; (d) undertaking checks to enhance data quality during data analysis such as data cleaning to remove outliers; (e) comparing data points reported from multiple citizens’ instruments in the same geographic location (postcode) to identify the degree of agreement amongst them. The more accurate the data, the more likely those points should overlap; (f) comparing citizens’ recordings to the official weather data for a specific area and identifying the degree of agreement or distance between the recordings. We discuss how such good practice guidelines can be leveraged to improve data quality for air quality measurements in Section 4.

To the best of our knowledge, none of the surveyed IoT-CS systems provides sufficient evidence that they thoroughly meet these challenges to attain sensor measurements so that they are of a high quality similar to those of standardised, calibrated, scientific instruments. This agrees with the findings of others, e.g., [29]. Hence, rather than having an IoT-CS project goal for citizens to be able to undertake high-quality measurements using LCS in variable operating conditions to determine the environmental quality, e.g., the AQ level according to a standard, we focus on their ability to satisficing as a goal.

To answer RQ1, our survey of related work indicates the multiple dimensions and complexity of sensing needed to better understand the various factors that affect IoT-CS data quality of AQ monitoring. In this pilot study, we also adopted good practice guidelines to promote higher data quality in CS studies in practice as proposed by [24]. Additionally, we also proposed a classification of CS instruments to help better understand the data quality challenges when using different types of CS scientific instruments.

## 3. LCS IoT-CS AQ Study in 4 Cities

A pilot study for a project called Scientific Analytics of Green Everyday Internet of Chemistry Things (SAGE-IoCT) was researched, developed and then evaluated as part of a wider funded initiative called the UK Global Challenges Research Fund that is in part related to the United Nations Sustainability Development Goals (SDG) for 2030, in particular, Goal 11: Sustainable Cities and Communities [30]. Based upon the analysis of related work for IoT-CS projects, in the pilot study, we could not support citizens to undertake standardised AQ measurements to relate to the standard AQI and thus classify their AQ to standard levels from good to hazardous. This was mainly due to practical constraints to check our AQ LCS against calibrated instruments and the otherwise need for citizens to take AQ measurements over longer periods to get a long-enough averaged value according to the standard. First, the study goals were defined in terms of what to satisfice:*Raising awareness of temporal-spatial variations* in the environment state in the habitat close to where someone lives or works that has meaning to them, e.g., that they could observe high versus low temporal-spatial variations in AQ using an AQ LCS IoT kit;*Determining if increasing proximity to artificial sources of pollution* decreases the environmental quality;*Providing some preliminary field evidence that there may be some citizen-driven issues* and concerns about their local environmental state warranting the need for additional, more rigorous standardised measurements to confirm or refute the IoT-CS LCS measurements.

Second, the method and process to achieve the goals were defined. Third, the study was undertaken, and then its results were evaluated. This pilot project had a limited budget of £14k and a time-span of six months. Although we accept that user engagement and user experience are important [31], it was not the focus of our project, given that we only had access to limited resources to support this pilot project. We recruited 21 participants, gender-balanced, who were students at Pakistan universities. In terms of the DITO (Doing It Together Science) project’s escalator framework for public engagement [31], we position our participants at Level 6, as this requires regular data collection and analysis by participants.

### 3.1. Apparatus

This kit was based upon a low-cost Arduino Uno MCU board for local sensor data acquisition, extended with a Wi-Fi extension board called a Wi-Fi shield to exchange the data with a data server and a sensor shield to ease the connection of multiple AQ sensors, plus a battery power pack to power all these (for 5 h or so). The gas sensors included: a dust sensor (Grove PPD42NS) to measure Particle Matter (PM)—size not specified, but the sensor is responsive down to 1 μM; a combined CO and NO_2_ (Grove Multichannel Gas) Sensor and an SO_2_ (Logele G123) sensor. An overview of the architecture of the kits is shown in Figure 2. 20 of these IoT CS AQ kits were assembled and tested in terms of responsiveness. They reacted as expected to short artificial stimuli, such as burning a candle or spraying air fresheners or perfume nearby. Such stimuli experiments were also proposed at the workshop such that participants could replicate these after the workshop (see Figure 3, Figure 4 and Figure 5). These readings (if the sensors were calibrated) indicate that the air quality when spraying perfume is hazardous for a short time. This agrees with other studies that indicate perfume sprays contain VOCs that are considered harmful [32]. Note also that VOC measurements are not defined within standard AQIs. This may be because VOCs may occur more problematically indoors, whereas most standard AQ instruments are situated outdoors. Harmful VOCs may have low concentrations, and detrimental health symptoms may be slow to develop.

### 3.2. The Workshop

Next, citizens were recruited and invited to attend a half-day IoT-CS AQ monitoring workshop. This citizen science project involved volunteers that were mainly students but not experts in environmental science who showed some common interest concerning the need to monitor air quality because it also impacts human health. In terms of researchers directly leading the study, it was not a multi-disciplinary team but composed of faculty members who had a computer science engineering background rather than an environment science one.

At the workshop, participants received some background information about AQ monitoring, e.g., to discuss sources of bad AQ indoors and outdoors as examples of where and what causes AQ changes. They also received hands-on training on how to use the IoT kit. We proposed that the main goal of the study was to satisfice that they could take AQ measurements in their surroundings and discover average, high and low spots of AQ readings, temporally and spatially. Participants were also asked to note, where possible, the sensor operational conditions such as location coordinates, time, temperature using additional devices and note the type of location (e.g., indoors, outdoors at a busy crossroads), and what, if any, human activities affected the AQ occurred in the vicinity. Participants took AQ measurements in four different cities in Pakistan, within one major region within each city: Islamabad (the H-12 region of the city where NUST is located, Rawalpindi (New Lalazar region), Taxila (the region where UET is located) and Wah (Cantonment region, often abbreviated to Wah Cantt).

In terms of the opportunistic versus participatory sensor data acquisition process we used in this pilot study, once citizens left the workshop, we adopted a hands-off approach to how citizens decided to collect the data, how they exchanged data with us (we set up no central public server), which AQ data they collected, in which physical world context they collected it, where they went when they collected the data and for how long they collected the data. We wanted to empower citizens to have the choice and control to collect the data on how, where, and when they wanted to.

The minimum time for data collection was 8 min, and the maximum was 2.5 h. Although we said we would not aim to present standardised measurements for the reasons given previously, we did use the AQI standards to help interpret the results even though our input data for analysis may not be of high data quality. Essentially, this meant that we made a relative comparison to compare different readings temporally and spatially, interpreting the values of our measurements concerning each other and concerning the standards even though there may be a response bias, non-linear response, offset errors etc., when using our LCS.

In response to RQ2, user engagement workshops such as the one described in this section can be an effective method of making citizen science more inclusive. The design, organization and content of these workshops can directly influence the interest and motivation of the attendees to participate in CS projects.

Each of the 21 attendees of the workshop was given a survey questionnaire. The workshop was initially assessed by the participants on a scale of 1 to 5, 1 being ‘insufficient’ and 5 being ‘excellent’. Of these, 62% of the participants assessed the workshop as ‘excellent’. Furthermore, 100% of the participants thought that the workshop met the program objectives, and 95% were confident about the knowledge and information that they had gained on air quality monitoring. Moreover, on a scale of 1 to 5, 50% rated their level of enjoyment working with the Arduino as 4 (Good), and 52% rated their experience of AQ monitoring as 5 (Excellent).

The volunteers were asked about their interest in taking part in a citizen science-based AQ monitoring project before the workshop and then were asked once again after the workshop. Before the workshop, 43% explicitly expressed their interest. The rest were somewhat interested in AQ monitoring or citizen science in general. After the workshop, this number increased to 78%, while the remaining volunteers expressed some interest in working on some kind of CS project in future.

The results of this survey imply that reaching out to citizens with the intent to impart information and to equip them with the basic skills and means would help to make science more inclusive. By giving opportunities to individuals with no (physical, geo) science background to take part in scientific activities of local interest, tasks such as data collection can be performed by ordinary citizens if proper training is given, and some motivation as to why to take part in CS is given.

### 3.3. Results

Some example results of the analysis that participants undertook to demonstrate differences in the air quality in four different cities of Pakistan in the time interval 1700–2100 h during the period from 19 March 2018 to 21 March 2018 are presented and discussed below. Outdoors, the air quality level in all four selected cities was hazardous for the same period (Figure 6).

Indoors, the air quality in Taxila and Rawalpindi was hazardous compared to that of Islamabad (Figure 7) during the same period. The major AQ pollution contaminants seem to be dust and CO. Figure 8 shows the AQI of the cities of Islamabad, Rawalpindi, Wah and Taxila for some overlapping periods between 6:30 p.m. and 11:00 p.m.

In response to RQ3, it is evident from Figure 6, Figure 7 and Figure 8 that satisficing/uncalibrated readings can provide some useful results. The AQI values in all three figures are within the normal range and provide a useful comparison of the cities in question. Though historical AQ data for all four cities was not available, Figure 7 shows the relative AQI that one would expect, with Islamabad being the city that experiences more wind and rain and less traffic than Rawalpindi and Wah Cantt should have a relatively lower AQI. When the AQI for Wah Cantt is compared versus Taxila (Figure 8), the AQI in Taxila significantly reduces from 6:30 to 10:30 p.m., while it seemed to remain constant in Wah Cantt over the same period. Our hypothesis here is that because UET/Taxila is not situated close to a main arterial road, unlike Wah Cantt, its traffic does drop as the evening wears on, leading to a drop in the AQI, unlike for Wah Cantt. Additionally, note the challenges in undertaking voluntary distributed CS in the wild. As this project had minimal resources, it could only supply one researcher to oversee the study. She was not able to oversee the collection of the data in four different cities by voluntary CSs at the same time. This led to data being collected over different periods in different cities despite directions to the contrary and made it difficult to compare AQI for Islamabad and Rawalpindi versus the other two cities. Performing CS with a low budget is especially challenging. Though the data collected in this study is not sufficient to make a perfect comparison, it does still satisfice in terms of identifying some useful insights concerning a comparison of the relative AQIs in these cities.

Moreover, the Pearson Correlation Coefficients based on N = 75748 data entries were calculated between each outdoor pollutant. The highest correlation was between CO and SO_2_ (r = 0.874), which is probably because both gasses are primary and secondary vehicular emissions, respectively. A relatively low correlation, yet significant (r = 0.17) was found between CO and NO_2_. Though NO_2_ is also a secondary vehicular emission, it reacts with oxygen in the presence of sunlight to produce ground-level ozone, ultimately reducing its overall presence. It was also observed that NO_2_ does not correlate with SO_2_ (r = 0.005). Dust had smaller but significant correlations with SO_2_ (r = 0.71), CO (r = 0.82) and NO_2_ (r = 0.44). This initial statistical analysis provides useful information about the presence of different pollutants and their interaction with one another.

To address RQ3, it is evident from the results that satisficing/uncalibrated readings can provide some useful results. The AQI values in Figure 6, Figure 7 and Figure 8 are within the accepted range, and hence, a useful comparison of these four cities’ AQ readings at any given time can be made. An underlying assumption by professional scientists is that any data supplied by citizens to be processed via their input (opportunistic sensing) should be of sufficient quality to be comparable and aggregable. This is even though the equipment used by citizens is often under more variable, less controllable, in the wild operating conditions and could be undertaken with more variable (lower?) quality scientific equipment, e.g., LCS IoT-CS equipment. Hence, it is proposed that at a base level at least, a more feasible objective is that citizens satisfice that relative AQ measurements can be made to identify temporal–spatial variations in periods and regions that they frequent that are meaningful to them.

### 3.4. Discussion: The Need for Deep Citizen Science

In most CS studies, such as this one, citizens behave more as data collectors data consumers. The participants apply and gain only a somewhat shallow knowledge of science, shallower citizen science. In these CS studies, even though there may be a focus on user studies where citizens also act as co-creators of the study [27,28], the study design often limits citizens’ ability to deepen their knowledge of science, to become better and more knowledgeable amateur citizen scientists. Fostering a deep citizen science approach, defined as enabling an ability to deepen citizens’ understanding and quality of science, is needed. This can help address the shortfall of a very small proportion of trained professional scientists in the general population. This can also help drive a more inclusive knowledge and application of science en masse for the greater good of the local community and society as a whole.

The primary goal of deep citizen science is where through citizens’ local application of science, e.g., during CS, they deepen their knowledge and quality of science. This can be facilitated through various means as follows. A deeper user engagement in CS can be modelled and fostered by considering the escalator framework for public engagement from [31]. Although there is a link between deeper, more frequent participation in CS at the top end of the escalator and science awareness, participants could still act more as (frequent) spectators rather than actively increasing their science awareness and knowledge. This is borne out by [6], who comments that many hours of volunteering may be spent to support the vast diversity of CS and crowdsourced projects where citizens contribute by mass observations, while they are not necessarily deeply involved epistemically. CS can help deepen the meanings, possibilities, and implications of cooperation, particularly in science [6]. Further, openness in science will lead us to deepen the approach of the commons leading to a knowledge commons where knowledge is considered to be a complex ecosystem that operates as a common, shared resource that is digitally represented and can be openly accessed [6].

The use of IoT-CS can help widen and deepen CS by widening the range of scientific measurements via the use of low-cost DIY instruments. This can be a means of engaging citizens in experiential and hands-on forms of learning, such as learning by doing. The relevance of the science can be deepened via a stronger user context—a focus and awareness of local, situated needs, conditions, and challenges. The data quality and its interpretation can be deepened via data satisficing to help detect local variations and anomalies and via automating the recording of the important measurement context. For example, for AQ, this context can include location and time and meteorological data such as temperature, pressure, humidity, wind and precipitation.

## 4. Conclusions

We classified CS studies not by the field of study but by the equipment used and focused on IoT-(driven) CS, i.e., CS undertaken with IoT equipment.

Three research questions (RQ) were posed in this study to advance CS: What factors affect CS and the interpretation of IoT-CS AQ data quality (RQ1)? How can we make science more inclusive by dealing with the lack of scientists, training and high-quality equipment (RQ2)? Can a lack of calibrated data readings yield otherwise useful results for IoT-CS AQ data analysis (RQ3)? We addressed these as follows.

To address RQ1, our survey of related work indicates the multiple dimensions and complexity needed to better understand the various factors that affect IoT-CS data quality. To answer RQ2, user engagement workshops such as the one described in Section 3.2, can be an effective method in making citizen science more inclusive that also train users to operate the IoT-CS AQ devices more understandably. To address RQ3, it is evident from the results (Section 3.3) that satisficing/uncalibrated readings can provide some useful results and interpretation. An interesting open question raised in future directions of [4] relevant to this article is who gets to decide what science consists of? Further to this, we can add who gets to decide what quality of science is sufficient to fulfil some CS aims. This question remains open.

### 4.1. Main Findings

In addition, our main findings were that:1.*IoT-CS can provide a versatile platform to experiment with multiple factors offering new types of CS that would not otherwise be feasible*. A moveable IoT-CS device means that experimentation can be situated local to citizens, fostering strong ownership. Incremental experimentation can then be used to provide a more relevant context to the citizens involved, e.g., to find an individual’s norm, to detect (normal or abnormal) variations about this, and to investigate local AQ change sources that affect the normal environmental state.2.IoT-CS can raise awareness and broaden participation in science through the utility of more cost-effective and affordable IoT.3.*A classification of CS instruments can help us to better understand the data quality challenges when using different types of CS scientific instruments*.4.*An analysis of the data shows that it has been possible to detect AQ differences and to make an initial comparison*. Hence, the analysis provides some insight into the indoor and outdoor AQ within a city and between cities. However, from a CS point of view, this knowledge could also help to highlight mitigating solutions, e.g., keeping windows closed and less use of extractor fans may significantly improve AQ during a high-dust season.5.*Adopting the good practice guidelines to promote higher data quality in CS studies* proposed by [28]: (a) We gave equipment operational instructions in an accessible language at workshops; (b) we did not provide technical details about different devices and their accuracy as it is far more complex to ascertain this for AQ data than temperature data, and in our study, we only used one type of device; (c) we asked citizens to report contextual information such as location, time but we needed to add more context info, such as temperature, humidity and a more fine-grained spatial context such as the spatial topology. However, for the latter, this could be very challenging for citizens to define in such a way that could be automated, fused and correlated; (d) we considered that removing outliers is not all that necessary since our goal is to ‘satisfice’, which was done. The AQIs calculated were all within the normal range, and though we have no reference, they were also in line with the state of each city whose data was available. Note also, given the size of the data, many outliers would have cancelled each other out; (e) we did not compare data points reported from multiple citizens’ instruments in the same geographic location (postcode) to identify the degree of agreement amongst them because in our pilot study we had insufficient citizens and instruments; (f) we did not compare citizens’ recordings to official AQ data for the reasons given in recommendation 2, see below.6.For IoT AQ use, the majority of studies focus on PM sized 1,2.5,10 microns and far less on the other AQ indices of CO, NO_2_, O_3_, SO_2_ determined in many regional AQ standards and regarded as hazardous to humans and other living things.7.For IoT AQ use, very few studies consider the cross-correlation of changes in individual AQ gases [14] as an additional cross-check of AQ changes to satisfy local AQ change hypotheses.

### 4.2. Recommendations

Our main recommendations are:1.*A deep citizen science approach (Section 3.4) should be fostered to help address the shortfall of a very small proportion of trained professional scientists in the general population to support a more inclusive knowledge and application of science en masse for the greater good*.2.*It may not be useful or even feasible to cross-check measurements from cheaper versus more expensive calibrated instrument sensors in situ in some regions/locations for some CS studies. Hence, data satisficing could be more of a focus if this occurs*. This is because, in some regions, calibrated instruments may not be deployed. For example, during our study period, there were no available calibrated instruments operating in Rawalpindi, Taxila and Wah. Even where calibrated instrument readings may be available, significant variations in measurements due to distance may occur for physical phenomena such as air quality, along with canyon effects. These may cause readings to be different between two instruments that are not exactly collocated at the same time. Additionally, note that for some physical phenomena, e.g., temperature, it can be far easier to check than for others, such as air quality.3.*Additional cross-checks that go beyond checking if co-located LCS and calibrated AQ measurements correlate under equivalent conditions should be leveraged* to consider if IoT CS AQ studies support local satisficing of expected AQ changes, such as: considering correlations: when measuring multiple AQ indices, how well these correlate to each other [14], temporal correlations, e.g., carbon-based transport use should cause outdoor peaks in the morning and early evening; checking the normal responsiveness of AQ sensors to well-known physical environment stimuli such as burning candles, using air freshener sprays or perfumes (indoors) [33], as these can help to identify potential faulty or very inaccurate sensors, etc.4.*Clearly state how similar the conditions and context are between LCS and calibrated instrument measurements and consider how to model any differences between these, e.g., spatial differences* [17].5.There is a *need for multi-disciplinary input of scientists, technologists, designers and participants through participatory design and thinking approaches* to be able to address the challenges of engagement and agency of empowerment to meet the opportunity of sustainability of CS for informed change.6.*Citizens often require re-orientation, a mindset change, to improve their scientific undertakings*, i.e., to switch from the plug and play expectation when interacting with the digital world to a longer setup and operation time, constrained operation and to realise the slower changes of the physical world.7.With engagement, *a more local approach is needed to mitigate the influence of hazardous findings that are found in more technical research articles so that they may be converted to everyday activities*.8.*To make the meaning of CS findings more accessible and understandable to the more general public, both enhanced data and knowledge rendering techniques are needed*.9.For measurements to benefit local citizens, *there is a need to develop techniques that map to local satisficing CS criteria, highlighting local change factors*.

## Figures and Tables

**Figure 1 sensors-22-03196-f001:**
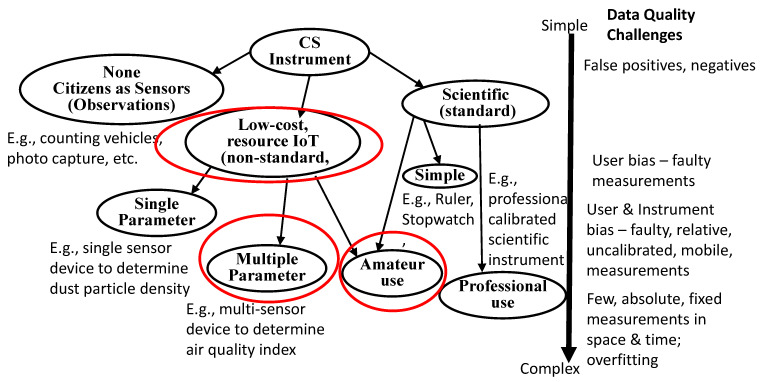
Taxonomy of CS Instruments (red circles indicate the focus of this AQ study).

**Figure 2 sensors-22-03196-f002:**
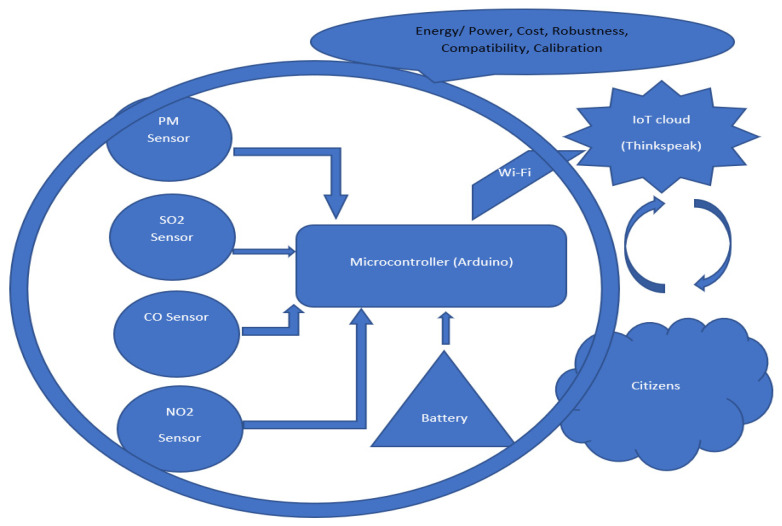
Architecture overview of the IoT kits designed for AQ monitoring.

**Figure 3 sensors-22-03196-f003:**
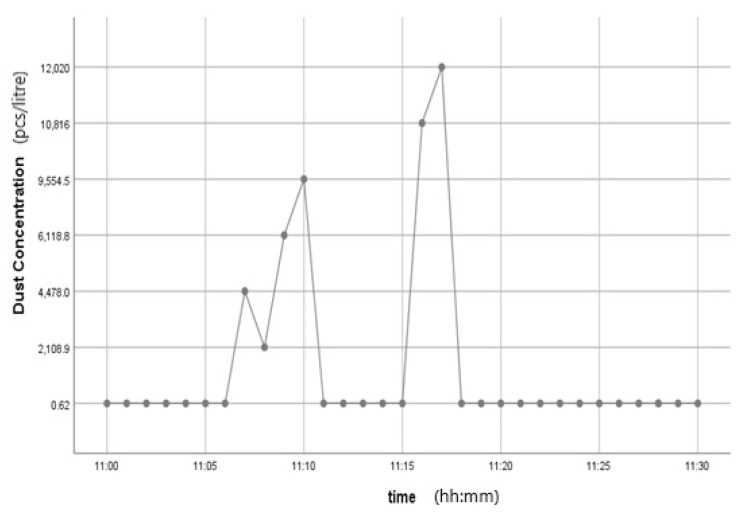
Time series graph showing the Dust PM variation after perfume was sprayed.

**Figure 4 sensors-22-03196-f004:**
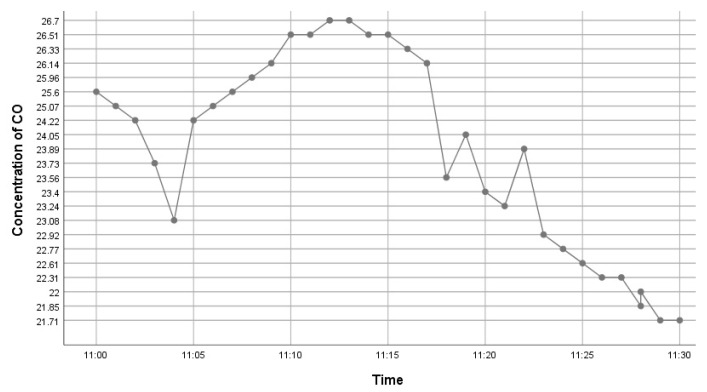
Time series graph showing CO variation after perfume was sprayed.

**Figure 5 sensors-22-03196-f005:**
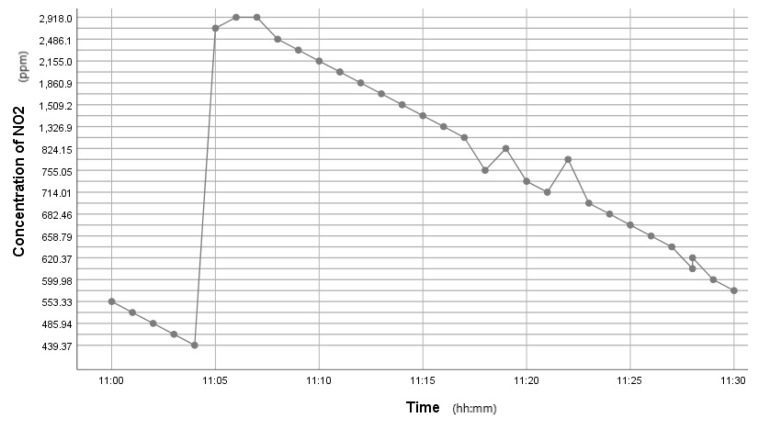
Time series graph showing the NO_2_ variation after perfume was sprayed.

**Figure 6 sensors-22-03196-f006:**
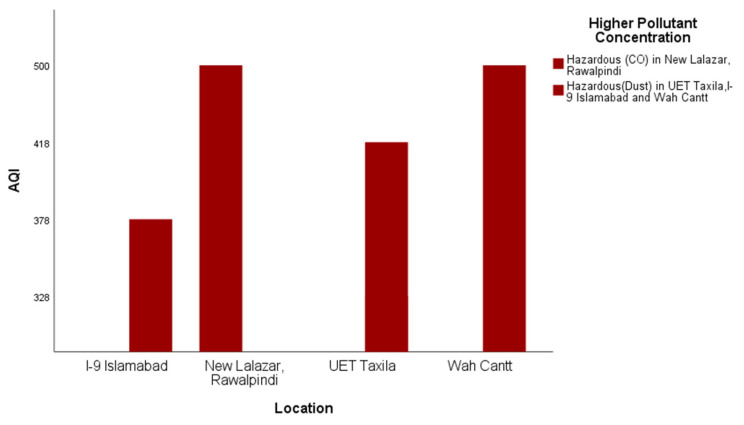
Outdoor AQI comparison in Rawalpindi, Islamabad, Taxila and Wah.

**Figure 7 sensors-22-03196-f007:**
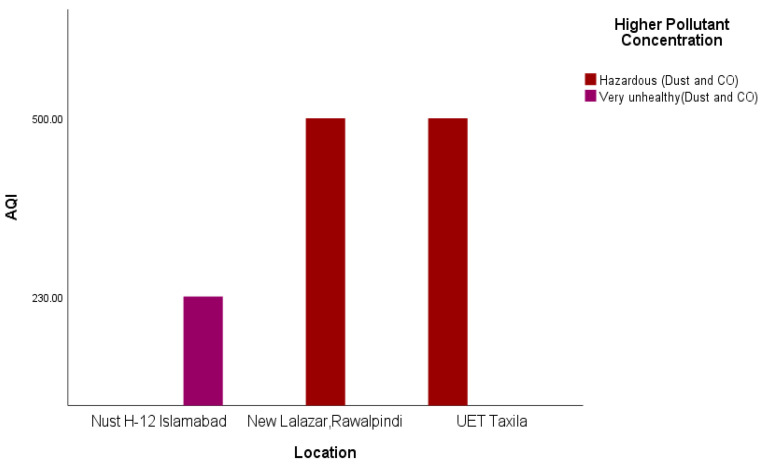
Indoor AQI comparison in Rawalpindi, Islamabad, and Taxila.

**Figure 8 sensors-22-03196-f008:**
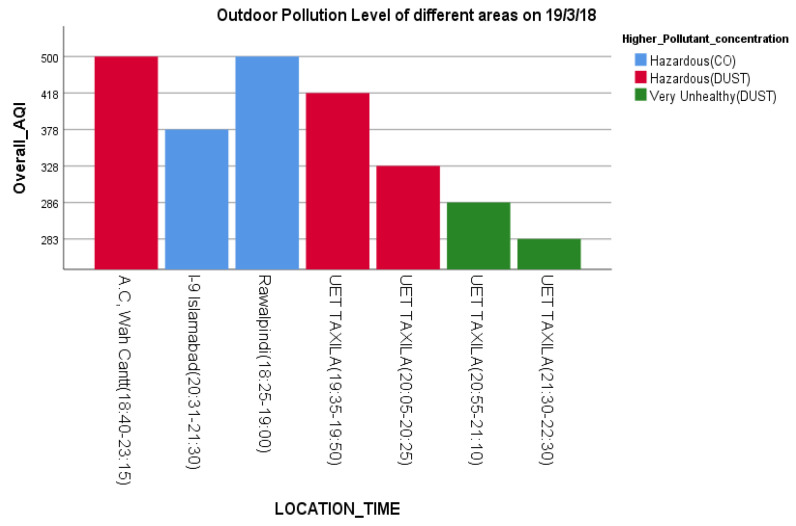
Indoor AQI Comparison of four different cities at different times.

## Data Availability

The data we acquired and processed is not made public but could be supplied if you contact the corresponding author.

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
