# Peer review of "How IoT-Driven Citizen Science Coupled with Data Satisficing Can Promote Deep Citizen Science"

_sensors, 2022, doi:10.3390/s22093196_

Round 1

Reviewer 1 Report

Please see the attached report.

Author Response

Reviewer 1

The authors set up and undertook a citizen science experiment for air pollution. The study is interesting but there are some lacks about the concepts of citizen science and the literature. Especially, in the last few years, several papers on the citizen science and the data quality have been published. I strongly recommend the following papers (but not limited with the following studies). These papers will enhance the readability of the manuscript and the conclusions.

  1. https://jcom.sissa.it/archive/20/06/JCOM_2006_2021_A02

  2. https://journals.plos.org/plosbiology/article?id=10.1371/journal.pbio.1002280

  3. https://jcom.sissa.it/archive/18/01/JCOM_1801_2019_E

  4. https://jcom.sissa.it/archive/18/01/JCOM_1801_2019_A02

  5. https://onlinelibrary.wiley.com/doi/10.1002/sce.21501

  6. https://wires.onlinelibrary.wiley.com/doi/10.1002/wat2.1578

  7. https://jcom.sissa.it/archive/19/06/JCOM_1906_2020_A03

  8. https://www.sciendo.com/article/10.2478/mgr-2019-0020

  9. https://onlinelibrary.wiley.com/doi/10.1111/aec.13062

  10. https://www.sciencedirect.com/science/article/pii/S0006320716303020?via%3Dihub

As can seen from the publications given above, the importance of CitSci is increasing and its application areas are widening. For this reason, the authors should give a proper literature review. In addition, the stages of a CitSci should be described and the position of the citizens involved in the project should be described clearly.

As a last comment, a conclusion section must be added and the conclusions obtained from the study must be highlighted and the abstract should be organized considering the conclusions.

Response: We made even clearer that the focus of this paper, is predominantly on CitSci combined with IoT applied to the AQ (Air quality) domain, rather than on CS in general, we nevertheless highlight some other CitSci work that comments on equipment and data quality issues, which is also our focus. Hence, we exclude some of the references proposed for CitSci. about non AQ domains such as fresh water, geography, ecology and conservation. Some others we added. Although we accept that user engagement and user experience is important, it’s not the focus for our project and given that we only had access to limited resources to support this pilot project. We added a statement about the stages of CitSci and the position of our students within it based upon the elevator model taken from ref. 1.

We thought that the heading main findings would be clear as a synonym for conclusions. We nevertheless re-termed our end section as conclusions. We also rewrote part of the conclusions and we rewrote the abstract to focus more clearly on the conclusions.

These are the refs. referred to

1. Skarlatidou A and Haklay M. Citizen science impact pathways for a positive contribution to public participation in science. Vol. 20, No. 06, 2021. J. Sci comm.
2. Wang, Y., Kaplan, N., Newman, G. and Scarpino, R., 2015. CitSci. org: A new model for managing, documenting, and sharing citizen science data. PLoS biology, 13(10), p.e1002280

User experience of digital technologies in citizen science

3. Skarlatidou, A., Ponti, M., Sprinks, J., Nold, C., Haklay, M. and Kanjo, E., 2019. User experience of digital technologies in citizen science. Journal of Science Communication, 18(01).

4. Skarlatidou, A., Hamilton, A., Vitos, M. and Haklay, M., 2019. What do volunteers want from citizen science technologies? A systematic literature review and best practice guidelines. JCOM: Journal of Science Communication, 18(1).

5. Phillips, T.B., Ballard, H.L., Lewenstein, B.V. and Bonney, R., 2019. Engagement in science through citizen science: Moving beyond data collection. Science Education, 103(3), pp.665-690

6. Metcalfe, A.N., Kennedy, T.A., Mendez, G.A. and Muehlbauer, J.D., 2022. Applied citizen science in freshwater research. Wiley Interdisciplinary Reviews: Water, p.e1578.

7. Golumbic Y, Baram-Tsabari A, Fishbain B. Engagement styles in an environmental citizen science project. Journal of science communication. 2020 Nov 4;19(6):A03.

8. Trojan, J., Schade, S., Lemmens, R. and Frantál, B., 2019. Citizen science as a new approach in Geography and beyond: Review and reflections. Moravian Geographical Reports, 27(4), pp.254-264.

9. Hall, M.J., Martin, J.M., Burns, A.L. and Hochuli, D.F., 2021. Ecological insights into a charismatic bird using different citizen science approaches. Austral Ecology, 46(8), pp.1255-1265. N/A on open access.

10. Gray, S., Jordan, R., Crall, A., Newman, G., Hmelo-Silver, C., Huang, J., Novak, W., Mellor, D., Frensley, T., Prysby, M. and Singer, A., 2017. Combining participatory modelling and citizen science to support volunteer conservation action. Biological conservation, 208, pp.76-86.

Reviewer 2 Report

I advise the authors to consider addressing the following comments:

  1. It should be ensured that words included in the title are not iterated as keywords; this could increase the discoverability of the paper once it is published.  
  2. In line 46, the number of a reference is [0] must be a typographical mistake and should be corrected.
  3. The sentence in lines 50-52 must be rephrased while authors should avoid using the term ecology type.
  4. In line 390, the authors state some motivations that should be provided. Could the authors state motivations which consider different groups of the public?
  5. The specific methodology which was used in this study must be justified by referring to specific literature sources.
  6. The paper needs to be edited by a professional English editing service as there are many grammatical mistakes throughout the text.

Author Response

It should be ensured that words included in the title are not iterated as keywords; this could increase the discoverability of the paper once it is published.

Response: we updated the keywords, Internet of Things (IoT), Citizen Science (CS); Data Quality; Data Satisficing

In line 46, the number of a reference is [0] must be a typographical mistake and should be corrected.

Response: this is reference [5] in our copy, we changed this.

The sentence in lines 50-52 must be rephrased while authors should avoid using the term ecology type.

Response: we remove the term ecology and referred more generally to some citizen science applications.

In line 390, the authors state some motivations that should be provided. Could the authors state motivations which consider different groups of the public?

Response: We add some specific motivation

The specific methodology which was used in this study must be justified by referring to specific literature sources.

Response: we’ve added some references suggested reviewer 1 to improve the justification of the methodology

The paper needs to be edited by a professional English editing service as there are many grammatical mistakes throughout the text.

Response: we put the paper through the Word spelling checker, the Grammarly checker & got 3 authors as native English speakers to further check.

Round 2

Reviewer 1 Report

Lack of a brief literature review on citizen science is still a bottleneck of the manuscript. However, this does not prevent to publish the manuscript. If this is provided, the readability of the manuscript could be increased. 

Author Response

Response to “Lack of a brief literature review on citizen science”. We feel our article already has such a brief review in the 1st paragraph of the introduction and in its following bullet points, supported by 5 references. Nevertheless, we improved and expanded on this further by adding 4 more references and some discussions of key points.